# The *MTNR1B* rs10830963 Variant in Interaction with Pre-Pregnancy BMI is a Pharmacogenetic Marker for the Initiation of Antenatal Insulin Therapy in Gestational Diabetes Mellitus

**DOI:** 10.3390/ijms19123734

**Published:** 2018-11-23

**Authors:** Gábor Firneisz, Klara Rosta, Zahra Al-Aissa, Orsolya Hadarits, Jürgen Harreiter, Ákos Nádasdi, Dagmar Bancher-Todesca, László Németh, Péter Igaz, János Rigó, István Sziller, Alexandra Kautzky-Willer, Anikó Somogyi

**Affiliations:** 12nd Department of Internal Medicine, Semmelweis University, H-1088 Budapest, Hungary; alaissazahra@yahoo.com (Z.A.-A.); nadasdi.akos@med.semmelweis-univ.hu (Á.N.); igaz.peter@med.semmelweis-univ.hu (P.I.); somogyi.aniko@med.semmelweis-univ.hu (A.S.); 2MTA-SE Molecular Medicine Research Group, Hungarian Academy of Sciences–Semmelweis University, H-1088 Budapest, Hungary; 3Department of Obstetrics and Gynecology, Medical University of Vienna, A-1090 Vienna, Austria; klara.rosta@meduniwien.ac.at (K.R.); dagmar.bancher-todesca@meduniwien.ac.at (D.B.-T.); 4Department of Obstetrics and Gynecology, Semmelweis University, H-1088 Budapest, Hungary; hadarits.orsolya@med.semmelweis-univ.hu (O.H.); rigo.janos@noi1.sote.hu (J.R.J.); 5Gender Medicine Unit, Division of Endocrinology and Metabolism, Department of Internal Medicine III, Medical University of Vienna, A-1090 Vienna, Austria; juergen.harreiter@meduniwien.ac.at (J.H.); alexandra.kautzky-willer@meduniwien.ac.at (A.K.-W.); 6Department of Probability Theory and Statistics, Eötvös Loránd University, H-1088 Budapest, Hungary; laszlo.nemeth.92@gmail.com; 7Department of Obstetrics and Gynecology, Szent Imre Teaching Hospital, H-1088 Budapest, Hungary; isziller@gmail.com

**Keywords:** gestational diabetes mellitus, GDM, insulin therapy, AIT, lifestyle, medical nutrition therapy, genetics, gene variant, *Melatonin Receptor 1B* gene, *MTNR1B*, rs10830963

## Abstract

The rs10830963 variant of the *Melatonin Receptor 1B* (*MTNR1B*) gene is associated with the development of gestational diabetes mellitus (GDM). We hypothesized that carrying the rs10830963/*G* risk allele had effect on antenatal insulin therapy (AIT) initiation in GDM in a body mass index (BMI)-dependent manner. *Design*: In this post hoc analysis the *MTNR1B* rs10830963 genotype and the clinical data of 211 Caucasian GDM patients were assessed. As a first step, a pre-pregnancy BMI threshold was determined where the effect of *MTNR1B* rs10830963/*G* allele carrying on AIT initiation was the most significant using logistic regression. Maternal age adjusted real-life odds ratios (OR) values were calculated. The chi-square test was also used to calculate the p value and 10.000 bootstrap simulations were performed in each case to re-assess the statistical power and the OR. Carrying the *MTNR1B* rs10830963/*G* allele increased the odds of AIT initiation (OR = 5.2, *p* = 0.02 [χ^2^ test], statistical power = 0.53) in GDM patients with pre-pregnancy BMI ≥ 29 kg/m^2^. The statistical power reached 0.77, when the pre-pregnancy BMI cutoff of 27 kg/m^2^ was used and the genetic effect on AIT initiation was still significant, but only using the logistic regression model. Carrying the *MTNR1B* rs10830963/*G* risk allele—in interaction with pre-pregnancy BMI—is likely be considered as a candidate pharmacogenetic marker of antenatal insulin therapy initiation and should be further assessed in precision medicine trials in GDM.

## 1. Introduction

Gestational diabetes mellitus (GDM) is characterized by a clinically relevant degree of hyperglycemia with onset at any time in pregnancy on the basis of certain adverse neonatal outcomes [1]. The International Diabetes Federation (IDF) estimates that one in seven births is affected by GDM [2] and this also poses a huge economic burden. The prevalence of GDM ranges from 8.1–14.8% in Hungary [3]. The costs of anti-diabetic medication, including insulin, the first-line agent recommended for GDM treatment in many countries (e.g., Hungary, Austria, USA) [4,5,6,7], are likely to increase [8]. The overall costs of antenatal insulin therapy (AIT) are influenced by the changing GDM diagnostic criteria also. The prevalence of GDM may substantially increase upon implementation of the International Association of the Diabetes and Pregnancy Study Groups (IADPSG) diagnostic criteria [9]. Despite the fact that the relative proportion of GDM patients requiring AIT may be lower using the newer diagnostic criteria compared to the 15–30% suggested earlier by American Diabetes Association (ADA) using the Carpenter–Coustan or National Diabetes Data Group criteria, the total number of GDM patients requiring AIT might still increase. [5,9,10]. Nevertheless, precise estimates about the proportion of GDM patients ultimately requiring AIT are still lacking and may depend on a number of factors that results in a large heterogeneity of AIT use rate in the clinical practice. Accordingly, AIT use rates in GDM may be as high as 54.9% even using the IADPSG diagnostic criteria as indicated by a recent study [9].

We have recently completed a case-control association study assessing the effect of 77 single nucleotide polymorphisms (SNPs) on GDM development in Hungary and Austria [11]. We reported a preponderant effect of the rs10830963/*G* risk allele of the *Melatonin Receptor 1B* (*MTNR1B*) gene on GDM development [11] (odds ratios [OR] = 1.84/1.64 [95% CI:1.54–2.21/1.38–1.97] depending on the GDM diagnostic criteria [IADPSG/m99′WHO]). We found that the rs10830963/*G* allele occurs with a minor allele frequency (MAF) of 36% in our GDM population. The *MTNR1B* rs10830963 was also the gene variant most significantly associated both with the fasting and with the 2 h postchallenge plasma glucose (PG) levels at the routine oral glucose tolerance test (OGTT) performed between the 24th–28th gestational weeks (gw) (mean effect sizes fasting plasma glucose (FPG)/2 h PG: 0.21 mmol/L [95% CI: 0.11–0.30]/0.61 mmol/L [95% CI: 0.32–0.89] increases, *p* < 5 × 10^−4^/5 × 10^−4^) out of the 77 genetic variants assessed in the original study [11].

A potential interaction between the pre-pregnancy body mass index (BMI) and the *MTNR1B* rs10830963/*G* risk allele was suggested previously by a meta-analysis of case-control genetic studies, which indicated that the effect of carrying the *MTNR1B* rs10830963/*G* allele became significant on GDM development only in studies with patients with mean pre-pregnancy BMI values above 25 kg/m^2^ [12].

However, no prior report was found about the effect of the *MTNR1B* rs108309639 gene variant on the odds of the initiation of pharmacological treatment, including antenatal insulin therapy (AIT) in GDM.

The aim of this post hoc analysis was to assess, whether carrying the *MTNR1B* rs10830963/*G* risk allele has any effect on the initiation of AIT treatment modality in GDM in a pre-pregnancy BMI-dependent manner.

## 2. Results

### 2.1. Clinical Characteristics

Data of 211 Hungarian GDM patients were assessed in this post hoc analysis. The major maternal clinical characteristics of the GDM populations, including PG values at standard 75 g OGTT, HbA_1c_ levels, maternal age, pre-pregnancy BMI, and weight gain are indicated in Table 1 according to pre-pregnancy BMI group, and the treatment modality.

In general, the GDM management in the study might be considered appropriate based on the HbA_1c_ values (95% CI: 5.04%–5.59% [32 mmol/mol–38 mmol/mol] in the different study groups, respectively). Nevertheless, the 3rd trimester HbA_1c_ values were only measured in 53.1% of the Hungarian patients (Table 1) consistent with our guidelines, which did not routinely recommend the use of HbA_1c_ in GDM [4] due to the lack of internationally accepted cut-off values and to the altered red blood cell (RBC) turnover in pregnancy.

We report the clinical characteristics of the GDM population studied in pre-pregnancy BMI stratified subgroups to provide a detailed clinical description for the GDM subpopulation where our genetic findings are the most relevant. Therefore, we use the pre-pregnancy BMI cutoff value of ≥29 kg/m^2^ due to the fact that the logistic regression model indicated the most significant genetic effect on AIT initiation at this cutoff (see also under Section 2.2.1).

GDM patients with pre-pregnancy BMI ≥ 29 kg/m^2^ were treated significantly more often with AIT (25%) (10.2%, *p* = 0.029); had significantly higher FPG (*p* = 1.4 × 10^−5^) levels and gained less weight during pregnancy (*p* < 0.0001), compared to those with lower BMI values. Higher FPG levels were found in the insulin treated compared to the non-pharmacologically treated patients within both pre-pregnancy BMI defined GDM subgroups (*p* = 0.0249 and *p* = 0.0116, higher and lower BMI groups, respectively) (Table 1). The AIT rates were 25.8% vs. 10.1% in the BMI defined subgroups (higher than or equal to vs lower than 29 kg/m^2^, respectively, Table 1). The weight gain of GDM patients with pre-pregnancy BMI ≥ 29 kg/m^2^ was lower in comparison to those with lower pre-pregnancy BMI values.

### 2.2. Genetic Effect on Antenatal Insulin Therapy (AIT) Initiation–Body Mass Index (BMI) Dependency

#### 2.2.1. Genetic Effect on AIT Initiation–BMI Dependency

An interaction was detected between pre-pregnancy BMI and carrying the *MTNR1B* rs10830963/*G* allele on the odds of AIT initiation in Hungarian patients with GDM.

The logistic regression model using a step-wise increase of pre-pregnancy BMI thresholds indicated a significant genetic effect on AIT initiation in GDM patients with pre-pregnancy BMI ≥ 27 kg/m^2^ (logistic regression: OR = 2.34, *p* = 0.037). This effect became the most significant when the pre-pregnancy BMI was higher than or equal to 29 kg/m^2^ (logistic regression: OR = 5.55, *p* = 4 × 10^−3^, Table 2). Subsequent calculation at the most significant pre-pregnancy BMI threshold using a contingency table and χ^2^ test resulted in the following OR and *p* values, respectively: Carrying the *G* risk allele of the *MTNR1B* rs10830963 was associated with a significantly higher odds of AIT initiation in patients with GDM and pre-pregnancy BMI ≥ 29 kg/m^2^ (OR = 5.2, *p* = 0.02 [χ^2^ test], statistical power = 0.53, Table 3). In contrast, when we used the BMI ≥ 27 kg/m^2^ as a pre-pregnancy BMI threshold, the chi-square test was non-significant (OR = 2.15, *p* = 0.18).

The *MTNR1B* rs10830963 genotype distributions and the number of GDM subjects in the AIT/non-AIT treatment modality subgroups according to the pre-pregnancy BMI cutoffs of 27 kg/m^2^ and 29 kg/m^2^ are indicated in Appendix A.

### 2.3. Genetic Effect in Combination with Glycemic Traits, Internal Validation of Prediction Model Using Bootstrap Method, Statistical Power

#### 2.3.1. Genetic Effect in Combination with Glycemic Traits

We could neither detect any effect of genotype on the total daily insulin dose required nor any significant genetic effect on the initiation time of AIT. Regarding the FPG levels (at OGTT) our results indicated that the effect of carrying the *MTNR1B* rs10830963 *G* allele on AIT initiation is substantially weakened to a statistically NS degree (OR = 2.54, *p* = 0.06). FPG was validated as a previously known predictor of AIT for the entire GDM population (OR = 2.43, *p* =3 × 10^−4^) confirming that FPG levels are true mediator variables in the entire GDM population and therefore should not be included in this analysis. However when both genetics and FPG levels together were assessed as predictors of AIT initiation in pre-pregnancy BMI-defined subgroups, we found that the genetic effect was not significant and the FPG levels became a significant predictor (OR = 2.34, *p* = 0.01, maternal age adjustment) in the lower pre-pregnancy BMI (<29 kg/m^2^) subgroup; in contrast to the higher BMI (≥29 kg/m^2^) subgroup, where the *MTNR1B* rs10830963 *G* allele carrying had significant effect (OR = 14.15, *p* = 0.01) on AIT initiation and the FPG effects were statistically non-significant.

When the 120 min PG values at OGTT were included in the analysis the genetic effect became NS, indicating that the 120 min PG values should be excluded from the analysis of the entire population. Subsequently we found that the 120 min PG at OGTT values tended to be associated with AIT initiation (OR = 1.49, *p* = 0.09) in the lower BMI group and conversely in the higher BMI subgroup the *MTNR1B* rs10830963 *G* allele carrying was significantly associated with the initiation of AIT (OR = 7.6, *p* = 0.02) in the logistic regression analysis.

#### 2.3.2. Internal Validation of Prediction Model Using Bootstrap, Statistical Power

Statistical power was re-calculated employing 10.000 bootstrap simulations in each case and resulted in a statistical power of 0.49 at pre-pregnancy BMI ≥ 29 kg/m^2^ with a mean OR of 7.9 for the genetic effect on AIT initiation. In order to reach a power of 0.8 and 0.95 the number of GDM patients needed to be enrolled was 125 and 200 in the BMI ≥ 29 kg/m^2^ subgroup, respectively. Still, when we applied the pre-pregnancy BMI cutoff ≥ 27 kg/m^2^ (maternal age adjusted *p* = 0.037 in the logistic regression analysis) which provides a larger subgroup of GDM patients the statistical power was 0.77 with a mean OR of 4.29.

## 3. Discussion

### 3.1. Potential Explanations for the Association of MTNR1B rs10830963/G Risk Allele Carrying and AIT Initiation in Gestational Diabetes Mellitus (GDM): The Role of β-Cell Dysfunction

The effect of the *MTNR1B* rs10830963/*G* risk allele on the initiation of insulin therapy in GDM was assessed in this post hoc analysis of a recently published study [11].

According to our knowledge these are the first data indicating that carrying a common gene variant has a significant effect on the initiation of insulin therapy in GDM. We found that the *MTNR1B* rs10830963/*G* risk allele was associated with a significant increase in the odds (OR = 5.2) of AIT initiation in Hungarian GDM patients with pre-pregnancy BMI ≥ 29 kg/m^2^ in real-life.

There is an increasingly better understanding of the role of β-cell dysfunction in the development of the most prevalent diabetes types, such as GDM and type 2 diabetes, frequently evolving on the basis of genetic susceptibility [13,14]. Twin studies suggested that the variability of insulin secretion is largely (i.e., 75%) heritable [15]. Prior reports confirmed that *MTNR1B* rs10830963 is a true causal gene variant in type 2 diabetes development [13] which impaired the early-phase insulin response [16,17] and also increased the odds of GDM development [11,12,18,19,20,21]. The *G* allele of rs10830963 confers risk of diabetes mellitus through a stage of isolated-IFG and a decrease of oral and intravenous glucose-stimulated insulin release, suggesting a pancreatic β-cell dysfunction and a decreased acute serum insulin response [22]. Some authors suggest that rs10830963 *G* risk allele increases islet *MTNR1B* expression via increased FOXA2-bound enhancer activity and NEUROD1 binding in islet cells [13], while others suggest that its effect on insulin secretion is secondary due to a central dysfunction [23].

### 3.2. Interaction Between the (Pre-Pregnancy) BMI and the MTNR1B rs10830963 Genetic Effects

Our hypothesis that the effect of the rs10830963/*G* allele on AIT initiation might occur only above a BMI threshold was consistent with the observation that increased odds of GDM development was reported only in studies including subjects with higher (>25 kg/m^2^) mean pre-pregnancy BMI [12]. Furthermore, a recent study, which applied early medical nutrition therapy (MNT) and lifestyle intervention from the 13th to the 24th–28th gestational week in high risk pregnant individuals (BMI > 30 kg/m^2^ and/or prior GDM in history) found that only non-carriers of the *MTNR1B* rs10830963 risk allele *G* benefited from the 3 months intervention as measured by the incidence of GDM diagnosed at routine OGTT [24]. Authors concluded that carrying the *MTNR1B* rs10830963/*G* risk variant could worsen the efficacy of early lifestyle and MNT interventions in high-risk pregnant individuals. Both the Finnish study [24] and our analysis assessed the effect of *MTNR1B* rs10830963 genotype in pregnant Caucasian women and had a primary outcome of reaching/exceeding the pre-defined glycemic targets (for GDM diagnosis/AIT initiation, respectively) after a period of MNT and lifestyle intervention in high risk pregnant/GDM individuals. Both the study [24] by Grotrenfelt et al. and our analysis consistently report genetic effect sizes characterized with high OR values (using the pre-pregnancy BMI cutoffs of 30 kg/m^2^ and 29 kg/m^2^, respectively). Recently, an additional clinical study suggested a role for the *MTNR1B* rs10830963 in gene–lifestyle interactions and GDM development risk in pregnant women [25].

It may be proposed that both pregnancy specific alterations and the implementation of a pre-pregnancy BMI threshold above which the genetic effect becomes the most significant may explain the difference found in the *MTNR1B* rs10830963 associated genetic effect sizes among type 2 diabetes mellitus (T2DM) [16,22,26] development; GDM development [11,20,27]; and AIT initiation in GDM reported here.

### 3.3. Interpretation of the Findings in Relation to Large Scale Studies on GDM Prevention in Ungenotyped Pregnant Populations with High Pre-Pregnancy BMI

The DALI study group reported that among pregnant women with a BMI of ≥29 kg/m^2^ who did not undergo genotyping, healthy eating and physical activity was unlikely to prevent GDM development nor it was a cost-effective early intervention to decrease the FPG and HOMA-IR values [28,29]. The DALI findings are potentially consistent with the interpretation that MNT and lifestyle interventions might only be beneficial in a subgroup of high-risk pregnant women, but not generally in an ungenotyped pregnant population despite the high pre-pregnancy BMI values.

### 3.4. Clinical Aspects of Anti-Diabetic Drug Therapy and the Need for Better Prediction of AIT in GDM

It may be recognized that there is a large heterogeneity in AIT use in GDM: ADA prior suggested that AIT was needed in 15–30% of GDM patients (under Carpenter–Coustan or National Diabetes Data Group criteria) and this rate may decrease provided that the IADPSG criteria are implemented [5]. Authors in Germany found a 37% relative increase in anti-diabetic agent use for GDM treatment from 2008 to 2012 [30] and AIT use rates in GDM may be as high as 54.9% even using the IADPSG (2013 WHO) diagnostic criteria according to a recent Portuguese study [9].

Metformin is increasingly used in the treatment of GDM, however its use in pregnancy is still controversial due to the lack of convincing long-term offspring safety data [31] and, therefore, insulin is the first-line agent in many countries, including the US also [7]. Even if the unresolved concerns with metformin use in pregnancy would have been cleared, it is important to recognize that 34.8% [32] to nearly half [33] of GDM patients who initiated their drug therapy with metformin eventually had to start AIT during pregnancy. These data imply a high need to find better predictors of AIT in GDM in addition to the traditional clinical risk factors such as the high pre-pregnancy BMI, the advanced maternal age and the earlier metformin introduction [32].

### 3.5. Summary

Our observation may be interpreted as follows: pregnancy is physiologically characterized by increased insulin resistance (IR) from the early 3rd trimester due to increased production of estrogen, prolactin, weight gain, increased adiposity that is physiologically accompanied by a β-cell adaptation in part induced by placental lactogen. The decreased insulin response in *MTNR1B* rs10830963/*G* allele carriers [17] could result in a gap between insulin need and insulin secretion. In GDM patients with pre-pregnancy BMI ≥ 29 kg/m^2^ carrying the rs10830963 risk *G* allele, this gap might become so substantial that the endogenous insulin secretion is no longer adequate to cover the increased insulin need despite the MNT and lifestyle intervention and the administration of exogenous insulin becomes necessary to provide the appropriate glycemic control.

In general, the FPG value at OGTT was one of the predictors of AIT in a validated model [34]. In this analysis we found that in the lower pre-pregnancy BMI subgroup the FPG level was a more significant predictor of AIT than the *MTNR1B* genotype; however, in the higher pre-pregnancy BMI subgroup the *MTNR1B* rs10830963 genotype became superior compared to the FPG. This indicates that the *MTNR1B* rs10830963 gene variant may have an additional direct effect on the initiation of AIT in the subgroup of GDM patients with pre-pregnancy BMI ≥ 29 kg/m^2^. In addition, from the clinical perspective it is notable that the intra-individual reproducibility of PG results at 75 g OGTT, including FPG, are under intense scrutiny (in the non-pregnant population) [35], in contrast to the genomic DNA information that is expectedly a stable marker.

Furthermore, it was independently [24] reported that carrying the *G* risk allele of *MTNR1B* rs10830963 significantly decreased the odds of responding to an early MNT and lifestyle intervention with a similar effect size in a high-risk pregnant population.

### 3.6. Significance of Findings

AIT is highly effective at providing good glycemic control, however it requires skilled handling, may lead to increased fear of therapy and additional weight gain during pregnancy, residual weight excess after delivery [36], is associated with increased risk of post-partum diabetes [37], poses a risk for hypoglycemia, and is costly.

The trends in modern societies and the more prevalent classical GDM risk factors, such as higher pre-pregnancy BMI [32,38] and higher maternal age [8,32] at pregnancy might underline the need for novel strategies in predicting the need for AIT, including the precision medicine approach to provide better care and also decrease the overall economic burden.

Nevertheless, the economic utility and clinical benefit of the precision medicine approach may only be accurately assessed in randomized controlled clinical trials (RCTs) and subsequent cost/benefit analyses. Due to the fact that this first report is only a single post hoc analysis of a prior case-control genetic study, no conclusion can be made regarding any potential economic utility and clinical benefit that the *MTNR1B* rs10830963 genotyping would mean in a high-risk GDM population.

### 3.7. Limitations

Although this first observation using the BMI cutoff of 29 kg/m^2^ obtained from the adjusted logistic regression model remained significant even in the χ square test, our study was limited by the sample size, the statistical power, the non-interventional nature of the post hoc analysis, and also by the lack of replication, and therefore should be interpreted with caution. However, it may be argued that the statistical power (using the bootstrap technique for the internal validation of the prediction model) reached 0.77, above the pre-pregnancy BMI cutoff of 27 kg/m^2^ and the genetic effect on AIT initiation was still significant, but only in the adjusted logistic regression model and not in the χ square test.

## 4. Subjects and Methods

### 4.1. Study Design, Participants, Inclusion and Exclusion Criteria, Initiation of AIT

#### 4.1.1. Study Design and Participants in the Original Study

After approval from the relevant institutional bodies (Semmelweis University, Regional and Institutional Committee of Science; the Medical Research Council Scientific and Research Committee of Hungary and the Research Ethics and Ethics Committee of the Medical University of Vienna) we have conducted the original study according to the declaration of Helsinki in two countries and three centers (one in Austria, two in Hungary) [11]. A completed and signed informed consent was obtained from each patient or subject after a full explanation of the purpose and nature of all procedures used. Pregnant women were enrolled in the case-control study on a voluntary participation basis in the framework of a European Foundation for the Study of Diabetes (EFSD) New Horizons Research Initiative between 2012 and 2015. The results of this association study with 77 gene variants in a central-European GDM population has been recently reported [11]. This post hoc analysis was subsequently performed to assess the effect of the *MTNR1B* rs10830963/*G* risk allele on the initiation of AIT in the Hungarian GDM population (Appendix A).

#### 4.1.2. Inclusion Criteria

The diagnosis of GDM in Hungary was established according to the guidelines of the Hungarian Diabetes Society using the modified 1999 World Health Organization (WHO) recommendation [GDM: 75 g OGTT at 24th–28th gw: FPG ≥ 6.1 mmol/L, 120 min PG ≥ 7.8 mmol/L] [39,40] that was in place between 2012 and 2015 when patients were enrolled to the original study [11]. All Hungarian GDM patients from the original study were included in this post hoc analysis of the genetic effect on AIT initiation, provided that *MTNR1B* rs10830963 genotype information and the clinical records on treatment modality were available.

#### 4.1.3. Exclusion Criteria

Exclusion criteria of the original study were based on the Strengthening the Reporting of observational (case-control) studies in Epidemiology statement for Genetic Association studies (STREGA) guidelines [41]. We also excluded patients with overt diabetes or other diabetes forms than GDM (e.g., type 1 DM) or chronic medication use with known impact on glycemic control, in vitro fertilization, twin pregnancies or variables likely to be associated with population stratification (e.g., non-European ancestry).

#### 4.1.4. Initiation of AIT

At the time of the decision about the initiation of AIT, both the doctors and the patients were blinded to the results of the *MTNR1B* rs10830963 genotyping. The genetic information remained hidden through the entire pregnancy in all cases [11]. Patients were allocated to AIT in the routine GDM care by their diabetes specialist doctors based on reaching the pre-defined glycemic targets indicated in the Hungarian guideline: AIT was recommended if at any of the two-weeks follow-up visits after the initiation of non-pharmacological (MNT, physical activity) treatment the following glycemic targets were not met: FPG: 5.3 mmol/L, 1 h: 7.0 mmol/L, 2 h: 6.7 mmol/L [4]. None of the patients received metformin therapy, insulin was the only approved pharmacological treatment for GDM management in Hungary, and none of the GDM patients in this study population refused the recommended AIT.

### 4.2. Clinical Data Collection, Genetic Data, Statistical Analysis

#### 4.2.1. Clinical Data Collection

We gathered relevant clinical data that was recorded earlier in the case-control study, including the PG levels at routine 75 g OGTT (24th–28th gw), HbA_1c_ levels, pre-pregnancy BMI, maternal age, birth-weight percentiles [11] and GDM treatment modality and performed a post hoc analysis in the Hungarian GDM population (Appendix A). 

#### 4.2.2. Genetic Data

*MTNR1B* rs10830963 genotype results were extracted from our EFSD GDM genotype database that was built up in the original study as described [11]. Briefly: after genomic DNA isolation using a magnetic bead based robotized approach (Hamilton Robotics, Magna Starlet, Bonaduz, Switzerland) from ethylenediaminetetraacetic acid (EDTA)-anticoagulated whole blood samples a fluorescence resonance energy transfer (FRET)-based competitive allele-specific polymerase chain reaction (PCR) (KASP™ LGC Genomics, Teddington, Middlesex, UK) was used for the bi-allelic discrimination in case of 77 common gene variants. All genotyping results, including the *MTNR1B* rs10830963 genotypes were extracted using the SNP2 viewer software (Version 4.0.0, Hoddesdon, UK) [11] and the clinical associations were assessed subsequently.

#### 4.2.3. Statistical Analysis

The genotype distribution of *MTNR1B* rs10830963 was non-deviant from the Hardy–Weinberg equilibrium (HWE) in the control group of the original study and we analyzed the data using the chi-square test under the dominant model as this genetic model lead to the most significant results for this SNP earlier [11].

The genetic effect on AIT initiation was studied separately in the GDM (case) population in different BMI subgroups using logistic regression. Effect sizes, primary OR and p-values were obtained from the logistic regression under the dominant genetic model using a step-wise increase of pre-pregnancy BMI thresholds to generate different pre-pregnancy BMI study groups in addition to the binary (*G* allele carrier or non-carrier) *MTNR1B* rs10830963 genotype data. All data were adjusted to maternal age. The pre-pregnancy BMI thresholds were determined with the step-wise increase: a-) where in the logistic regression model the genetic effect became significant and b-) a separate pre-pregnancy BMI threshold where it became the most significant. In the second step, we calculated the p-value using a chi-square test for independency of carrying the *MTNR1B* rs10830963 risk variant against AIT necessity using the BMI thresholds determined as above. OR values were calculated from the corresponding contingency tables. Statistical power was also determined. Genetic effect on the AIT initiation time (in gestational weeks) was assessed using a linear regression model. Clinical data distribution (FPG, 2hPG, HbA_1c_, weight gain, pre-pregnancy BMI, maternal age) has been assessed using the Shapiro–Wilks normality test. The Kruskal–Wallis analysis of variance (ANOVA) by ranks and the one-way ANOVA with Tukey′s post hoc tests were used for multiple comparisons.

We applied the “Type 1b” prediction model development following the transparent reporting of a multivariable prediction model for individual prognosis or diagnosis (TRIPOD) statement: the entire data set we could obtain was used for model development and an internal validation technique was applied. Internal validation techniques use only the original study sample and include methods as bootstrapping, which—according to the TRIPOD statement—is the preferred method to evaluate predictive performance over randomly splitting data into two groups (“Type 2a” analysis) [42]. Accordingly, bootstrapping was performed to assign measures of accuracy to sample estimates. The statistical power using bootstrap simulations (10.000 simulations in each case) was re-calculated as the estimation of the sampling distribution using random sampling methods was possible.

The role of PG levels at OGTT in relation to the genetic effect on AIT initiation was also assessed in order to discriminate whether PG levels should be viewed as a confounding or an intervening variable (“mediating variable”) in this analysis [43]. A variable that explains a relation and/or provides a causal link between other variables should be evaluated as an intervening variable and should not be included in the statistical analysis between the other two variables to avoid the problem referred as “over-adjustment” [44,45].

### 4.3. Data Availability

All the *MTNR1B* rs10830963 genotype distributions and the number of GDM subjects in the AIT/non-AIT treatment modality subgroups according to the pre-pregnancy BMI cutoffs of 27 kg/m^2^ and 29 kg/m^2^ are indicated in Appendix A as data underlying the findings.

## Figures and Tables

**Table 1 ijms-19-03734-t001:** Clinical characteristics and rates of antenatal insulin therapy (AIT) use in pregnant women with gestational diabetes mellitus (GDM) according to pre-pregnancy body mass index (BMI) defined subgroups.

Hungary (n = 211)							
Treatment Modality in Patient Subgroups According to BMI (kg/m^2^)	Plasma Glucose (PG) Values at 75 g Oral Glucose Tolerance Test (OGTT) on the 24–28th gw (mM)		HbA_1c_ %/in IFCC Unit: mmol/mol	Pre-Pregnancy BMI (kg/m^2^)	Age at Delivery (years)	Weight gain during Pregnancy (kg)	Proportion of Neonates LGA (%) ^6^
	0′	120′					
≥29 kg/m^2^ (n = 63)							
25% ^1^ on AIT (95% CI)	5.9 (5.36–6.43) ^2^	9.03 (7.83–10.24)	5.59 (5.21–5.98)/37.6 (33.4–41.9) ^4^	34.02 (31.95–36.08)	36 (34.26–37.74)	4.56 (1.57–7.55) ^5^	25.0%
75% on MNT and lifestlye modification (95%CI)	5.22 (4.96–5.48) ^2^	8.77 (8.32–9.22)	5.34 (5.20–5.49)/34.9 (33.3–36.5)	34.2 (32.76–35.65)	33.22 (31.80–34.64)	5.48 (3.48–7.48) ^5^	13.04%
<29 kg/m^2^ (n = 148)							
10.2%^1^ on AIT (95%CI)	5.37 (4.91–5.82) ^3^	9.2 (8.28–10.12)	5.15 (4.82–5.49)/32.8 (29.2–36.5)	24.09 (22.35–25.82)	35 (32.12–37.88)	9.36 (7.52–11.19) ^5^	6.67%
89.8% on MNT and lifestyle modification (95%CI)	4.75 (4.63–4.87) ^3^	8.67 (8.28–10.12)	5.04 (4.92–5.16)/31.6(30.3– 32.9) ^4^	23.4 (22.90–23.91)	33.8 (33.02–34.58)	10.04 (9.30–10.78) ^5^	5.22%

GDM patients with BMI ≥ 29 kg/m^2^ were treated significantly more often with AIT (*p* = 0.029 ^1^) and gained less weight during pregnancy (*p* < 0.0001 ^5^). Insulin treated patients had significantly higher fasting plasma glucose (FPG) values within both pre-pregnancy BMI subgroups (*p* = 0.0249 ^2^, *p* = 0,0116 ^3^). Significant difference in HbA_1c_ levels (*p* = 0.0017 ^4^). The chi-square statistic is 8.7069 (*p* = 0.033 ^6^) for LGA proportions among different subgroups. MNT: Medical nutrition therapy. LGA: Large for gestational age. IFCC: International Federation of Clinical Chemistry and Laboratory Medicine.

**Table 2 ijms-19-03734-t002:** The interaction between pre-pregnancy BMI and carrying the *MTNR1B* rs10830963/*G* allele on the odds of AIT initiation in patients with GDM (real-life treatment data, logistic regression model).

≥BMI (kg/m^2^)	Effect Size	OR	*p*-Value	Number of Patients with GDM
20	0.544	1.722	0.0707	193
21	0.626	1.870	0.0548	177
22	0.628	1.874	0.0605	160
23	0.627	1.871	0.0642	141
24	0.563	1.756	0.1072	126
25	0.694	2.001	0.0642	108
26	0.736	2.088	0.0550	104
**27**	**0.850**	**2.341**	**0.0369**	**89**
28	1.134	3.108	0.0155	73
**29**	**1.713**	**5.548**	**0.0040**	**63**
30	1.592	4.912	0.0076	54
31	1.801	6.057	0.0127	46
32	2.705	14.951	0.0198	33
33	3.298	27.052	0.0319	26
34	2.726	15.274	0.0526	21
35	2.166	8.726	0.0962	18

Effect sizes, odds ratios (OR) and *p*-values were obtained from the logistic regression under the dominant genetic model using a step-wise increase of pre-pregnancy BMI thresholds to generate different pre-pregnancy BMI study groups, where BMI was used as a continuous variable thereafter in addition to the binary *MTNR1B* genotype data. All data were adjusted to maternal age. The logistic regression model indicated significant genetic effect on AIT initiation odds in GDM patients with pre-pregnancy BMI ≥ 27 kg/m^2^ and the genetic effect in the model became the most significant in patients with pre-pregnancy BMI ≥ 29 kg/m^2^. These pre-pregnancy BMI cutoff thresholds are indicated in **bold**. There is a non-linear increase in OR values by using a step-by-step pre-pregnancy BMI cutoff threshold increase.

**Table 3 ijms-19-03734-t003:** The effect of carrying the *MTNR1B* rs10830963/*G* risk allele on AIT initiation in GDM patients with above/equal or below the pre-pregnancy BMI threshold of 29 kg/m^2^.

Hungary (n = 211)	Odds Ratios of *MTNR1B* rs10830963/*G* Carriers for AIT Initiation Real Life AIT Set	*p* Value (Chi-Square Test)	OR 95% CI	Risk (*G*) Allele Frequencies	
				AIT	Non-pharmacological therapy only
BMI ≥ 29 kg/m^2^	5.2	0.02	1.3–20.8	53.13%	26.04%
BMI < 29 kg/m^2^	1.36	0.78	NA	40.00%	36.36%

Carrying the *G* risk allele of the *MTNR1B* rs10830963 was associated with higher odds of AIT initiation in patients with GDM and pre-pregnancy BMI above or equal to 29 kg/m^2^ (OR: 5.2, *p* = 0.02 [χ^2^ test]).

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
