# Peer review of "The *MTNR1B* rs10830963 Variant in Interaction with Pre-Pregnancy BMI is a Pharmacogenetic Marker for the Initiation of Antenatal Insulin Therapy in Gestational Diabetes Mellitus"

_ijms, 2018, doi:10.3390/ijms19123734_

Round 1
Reviewer 1 Report
1. First of all, this reviewer could not get the whole story of this paper due to the poor English, especially in introduction and discussion
2. In the result part, the flow chart of the participants would give us a better understanding of this study design. Especially, the indication of insulin treatment is not clear. The authors have stated that AIT was recommended if glycemic targets were not met after two weeks of non-pharmacological treatment. How was the portion of the patients who were agree to receive AIT?
3. Please clarify the reason why you have chosen BMI≥29 as the threshold instead of BMI≥27. Since the genetic effect on AIT initiation becomes significant from BMI≥27, I would recommend defining the differences between the data analyzed with BMI threshold of 27 and 29.
4. this reviewer could not understand the term “real-life AIT to LGA” or “LGA adjustment”. Were there indications of insulin treatment for those patients giving birth to LGA offspring? I cannot agree with the idea to include GDM patient giving birth to LGA in AIT group.
5. In the discussion part, function of melatonin becomes a topic all of a sudden. Although the authors have mentioned that increase in nighttime maternal melatonin levels was shown in this paper, there is no such data in result part. This paper only discusses about the gene variance of MTNR1B, no topic of melatonin function. Please show the data within the paper if the authors desire to discuss on this topic.
Author Response
Q1: First of all, this reviewer could not get the whole story of this paper due to the poor English, especially in introduction and discussion
A1: We have extensively revised the Manuscript which included an additional language editing and we do hope that this aspect has been also improved.
Q2: In the result part, the flow chart of the participants would give us a better understanding of this study design.
A2: We have constructed a flow chart, however due to this report is a post hoc analysis of data from a previously published study and not a prospective trial we think it should be indicated as a supplementary file. Please find the file uploaded.
Q3: Especially, the indication of insulin treatment is not clear. The authors have stated that AIT was recommended if glycemic targets were not met after two weeks of non-pharmacological treatment.
A3: The Hungarian guideline is specifically instructed us to do so. The glycemic targets are reported in the text and we have also referenced the Hungarian guideline (which is highly similar to other GDM management guidelines and largely based on them).
Therefore we indicated the translation of the guideline in the text as follows:
“AIT was recommended if at any of the two-weeks follow-up visits after the initiation of non-pharmacological (medical nutrition, physical activity) treatment the following glycemic targets were not met: FPG: 5.3 mmol/L, 1h: 7.0 mmol/L, 2h: 6.7 mmol/L [1].”
Q4: How was the portion of the patients who were agree to receive AIT?
A4: Although it may be somewhat unusual by Western standards, but we did not have any patient in this study in the Hungarian GDM population who would have had refused the recommended AIT. We have now included this information in the text as follows: “ None of the patients received metformin therapy and insulin was the only approved pharmacological treatment for GDM management in Hungary and none of the GDM patients in this study population refused the recommended AIT”
Q5: Please clarify the reason why you have chosen BMI≥29 as the threshold instead of BMI≥27. Since the genetic effect on AIT initiation becomes significant from BMI≥27, I would recommend defining the differences between the data analyzed with BMI threshold of 27 and 29.
A5: In our analysis we applied a logistic regression model to assess and explain the relationship between carrying the MTNR1B rs10830963 gene variant and AIT necessity in GDM.
We defined the high BMI subgroup by fixing a T threshold, the subgroup included GDM patients with pre-pregnancy BMI higher than T. By using different thresholds we could first observe significant association when T=27, however the genetic effect was the most significant, if we used the T=29 threshold, (see Table 2).
Subsequently after the logistic regression model, we applied a Chi-square test for AIT necessity and genetic variant to indicate the dependence.
When the T=29 threshold was applied, the chi-square test resulted in an OR=5.2 with p=0.02, which is a significant association. In contrast when we used the T=27 as BMI threshold, the chi-square test resulted in an OR=2.15 with p=0.18.
The difference between the results from the logistic regression analysis and Chi square test originates from the:
a-) adjustment (for maternal age) which is only applicable in logistic regression model
b-) the mathematical difference between the two approaches.
With these findings we concluded that the pre-pregnancy BMI³29kg/m2 subgroup is more suitable to report the MTNR1B genotype associated effect on AIT initiation at least with our sample sizes. However an extended analysis might be able to potentially identify the effect in a larger subgroup (e.g. BMI³27kg/m2 subgroup).
Q6: this reviewer could not understand the term “real-life AIT to LGA” or “LGA adjustment”. Were there indications of insulin treatment for those patients giving birth to LGA offspring? I cannot agree with the idea to include GDM patient giving birth to LGA in AIT group.
A6: There were no indications (at least not SMBG diary or other documented data) of AIT initiation in the “virtually insulin treated” group, where those GDM patients were allocated whom were without AIT, but an with LGA neonate delivery.
This was only an attempt to adjust our findings to biases that may occur in the daily life: SMBG measurements might be compromised without being clearly obvious for the doctor.
The decision about AIT initiation may be confounded by some factors too (Despite that our guideline for GDM management is “generally” strict, a certain flexibility is provided for the doctors in the assessment whether a patient reached the glycemic targets or not, e.g.: when a large number of SMBG readings are available for a period and only a few of them is above the target).
We thought that these points might be – at least in theory - potential biases and therefore we have done this separate analysis, when the “virtually” insulin treated subgroup was added to the real-life on AIT group. We felt reassuring that the genetic effect above the same BMI threshold on AIT initiation remained significant even using this (“LGA adjusted”) approach.
However, we admit that this extra analysis and approach was not fully justified as pointed out by the reviewer and may be haphazard by many factors including the observation that obesity even without GDM increases the odds of an LGA delivery.
According to the suggestion of the reviewer we fully removed the “LGA adjustment” from all sections (Methods/Results/Discussion) and report only the crude data obtained from GDM patients according to the treatment modality they received in real-life.
Q7: In the discussion part, function of melatonin becomes a topic all of a sudden. Although the authors have mentioned that increase in nighttime maternal melatonin levels was shown in this paper, there is no such data in result part. This paper only discusses about the gene variance of MTNR1B, no topic of melatonin function. Please show the data within the paper if the authors desire to discuss on this topic.
A7: We fully agree with the reviewer that we have not done circulating melatonin level determinations during day and night-time in our GDM study population.
The Japanese group did an outstanding work [2] in this highly demanding pregnant study population and proved that the night-time serum melatonin levels increased after 24 weeks of pregnancy, with significantly high levels after 32 weeks (3-4-fold increase! compared to non-pregnant) and these values decreased to non-pregnant levels on the 2nd day of puerperium. Even there was an increasing (NS) tendency in the day time serum melatonin concentration towards the end of pregnancy.
We thought that this was an interesting finding in pregnant women that would deserve its place in the discussion and may partly explain the unexpectedly large genetic effect sizes we report here.
However we must admit the Reviewer’s point as he/she may argue that these are Japanese findings [2] and the results are originating from an Asian population. In an epidemiologic study Chinese (men) had lower melatonin levels compared to Caucasian, African-American, and Ghanaian men, who had similar levels [3] therefore – on the analogy - we cannot fully rule out that unexpected differences may theoretically exist among pregnant populations with different ethnic origin.
Therefore – as requested by the Reviewer – we have taken out this aspect from the Discussion part of the manuscript.
We are grateful for the Reviewer for the positive comments and also for the critical questions and we do hope that our detailed answers point-by-point might be found satisfactory and this MS would be of interest to the Readers of the International Journal of Molecular Sciences.
References:
1. Hungarian Diabetes Association, The Diagnosis of Diabetes Mellitus, Adult Care of Patients with Diabetes Mellitus. Guideline. Diabetologica Hungarica 2011, XIX., (1 / Supplement 1. ), 62-65.
2. Nakamura, Y.; Tamura, H.; Kashida, S.; Takayama, H.; Yamagata, Y.; Karube, A.; Sugino, N.; Kato, H., Changes of serum melatonin level and its relationship to feto-placental unit during pregnancy. J Pineal Res 2001, 30, (1), 29-33.
3. Hsing, A. W.; Meyer, T. E.; Niwa, S.; Quraishi, S. M.; Chu, L. W., Measuring serum melatonin in epidemiologic studies. Cancer Epidemiol Biomarkers Prev 2010, 19, (4), 932-7.
Reviewer 2 Report
I read this manuscript with a great interest. The findings are intriguing and the conclusions are supported by the numbers presented. However, I have some minor concerns:
1. The lines 65-71
The costs of anti-GDM medication are likely to increase (lines 65-67). At the same time the number of GDM patients who need pharmacological therapy might be lower if the IADPSG diagnostic criteria are implemented (lines 67-71). The latter sentence suggests that the overall costs can decrease.
There's kind of contradiction that can be solved by a bit of rephrasing.
2. Clinical Characteristics
In general, it is unquestioned why subgroups of GDM patients were stratified according to the precise value of pre-pregnancy BMI 29 kg/m2. However, the reason for such a stratification should by clearly explained at the beginning of the article - in the text of the Clinical Characteristics. Possibly, the part of description of Figure 1 can be relocated to the main text and rephrased. Subsequently, the text below Figure 1 would become more concise and focused mainly on the p-values.
3. Figure 1
What is a number of cases >=29 kg/m2 and<29 kg/m2? Please, include these numbers.
4. Discussion
In general the discussion is well organized and the number of literature is adequate. I do agree that modern societies need the precision medicine approach to provide better care and also decrease the overall economic burden. However, in my opinion the MTNR1B rs10830963 gene variant assessment will not result in the decrease of the GDM-associated costs. On the contrary, these costs could highly increase, since females >=29 kg/m2:
1. should undergo genetic assessment
2. in case of positive result, should undergo active surveillance
3. should be treated finally with insulin.
Positive results the MTNR1B rs10830963 gene variant assessment will not change the modality of treatment, and will not have an impact on doses of insulin, tendency for hypoglycemia, and weight gain. The only benefit is an early identification of patients who need pharmacological therapy with insulin. Thus, the finding, despite its superior originality and great scientific significance, posses relatively low (i.e. lower than declared by Authors) practical and economical utility. Please, rephrase the discussion to avoid such preliminary conclusions.
Author Response
Q1: The lines 65-71
The costs of anti-GDM medication are likely to increase (lines 65-67). At the same time the number of GDM patients who need pharmacological therapy might be lower if the IADPSG diagnostic criteria are implemented (lines 67-71). The latter sentence suggests that the overall costs can decrease.
There's kind of contradiction that can be solved by a bit of rephrasing.
A1: We are grateful for this comment. We have rephrased and extended this section according to the suggestion of the Reviewer.
Q2: Clinical Characteristics
In general, it is unquestioned why subgroups of GDM patients were stratified according to the precise value of pre-pregnancy BMI 29 kg/m2. However, the reason for such a stratification should by clearly explained at the beginning of the article - in the text of the Clinical Characteristics. Possibly, the part of description of Figure 1 can be relocated to the main text and rephrased. Subsequently, the text below Figure 1 would become more concise and focused mainly on the p-values.
A2: We report the clinical characteristics of the GDM population studied in pre-pregnancy BMI stratified subgroups to provide a detailed clinical description for the GDM subpopulation where our genetic findings are the most relevant. Therefore we use the pre-pregnancy BMI cutoff value of ≥29 kg/m2 due to that the logistic regression model indicated the most significant genetic effect on AIT initiation at this cutoff.
We agree with the reviewer that this reasoning was not included in the „Clinical Charateristics” section, however it was only due to that first we had to find the most appropriate pre-pregnancy BMI threshold (T), which itself is rather to be reported in the „Genetic Effect on AIT Initiation - BMI Dependency” subchapter, which follows the „Clinical Charactersistics” in the Result section. We acknowledge that without any reasoning it may appear unexplained to the reader at first, therefore we followed the Reviewer’s advice and the revised text now includes a brief reasoning for the stratification of the GDM population to the pre-pregnancy BMI 29 kg/m2 threshold value in the Clinical Charateristics section also and not only a full explanation in the subsequent subchapter.
Q3: Figure 1
What is a number of cases >=29 kg/m2 and<29 kg/m2? Please, include these numbers.
A3: We have now included the number of cases on Figure 1.
Q4: Discussion
In general the discussion is well organized and the number of literature is adequate. I do agree that modern societies need the precision medicine approach to provide better care and also decrease the overall economic burden. However, in my opinion the MTNR1B rs10830963 gene variant assessment will not result in the decrease of the GDM-associated costs. On the contrary, these costs could highly increase, since females >=29 kg/m2:
1. should undergo genetic assessment
2. in case of positive result, should undergo active surveillance
3. should be treated finally with insulin.
Positive results the MTNR1B rs10830963 gene variant assessment will not change the modality of treatment, and will not have an impact on doses of insulin, tendency for hypoglycemia, and weight gain. The only benefit is an early identification of patients who need pharmacological therapy with insulin. Thus, the finding, despite its superior originality and great scientific significance, posses relatively low (i.e. lower than declared by Authors) practical and economical utility. Please, rephrase the discussion to avoid such preliminary conclusions.
A4: According to the points raised by the reviewer and rephrased this part of the Discussion.
We now state in the revised version that:
“the economic utility and the clinical benefit of the precision medicine approach may only be accurately assessed in randomized controlled clinical trials (RCTs) and subsequent cost/benefit analyses. Due to that this first report is only a single post hoc analysis of a prior case-control genetic study no conclusion can be made regarding any potential economic utility and clinical benefit that the MTNR1B rs10830963 genotyping would mean in a high risk GDM population.”
(In addition we think it is a highly interesting point and there may be even more interest in the MTNR1B Genotyping-GDM-Economic utility task that are not included in the discussion section of our MS but are as follows:
Recently, a Genome-wide association study (GWAS) of offspring birth weight in 86 577 women identified five novel loci and highlighted maternal genetic effects that are independent of fetal genetics [1]. Out of the 10 reported maternal genetic loci associated with offspring birth weight in a European ancestry meta-analysis of up to 86 577 women the highest genetic effect size was reported for the MTNR1B rs10830963 gene variant [1].
In the view that more than half of these samples were from the UK Biobank [1] it may be raised that this maternal genetic effect on offspring birth weight may also be the consequence of the undiagnosed GDM cases (and associated increase of neonates with macrosomia/LGA) in the UK maternal study population as there is no universal OGTT screening program for pregnant individuals in the UK.
It may be an interpretation, that as a result of non-universally preformed OGTT in the pregnant population, the proportion of undiagnosed GDM cases (and associated macrosomia/LGA deliveries) were potentially increased in the GWAS study population [1] and therefore this study (with no adjustment on maternal glycemia) eventually identified the MTNR1B rs10830963 variant as an independent maternal genetic factor associated with larger birth weights with the highest genetic effect size.
Since this UK GDM management approach is even supported by economic utility estimations [2], despite that the above remarks are only a conceptual interpretation to this recently published excellent GWAS study [1] these findings may still potentially argue that the decision making about the routinely performed OGTT in pregnancy might be difficult based solely on cost/benefit analysis [2] and may even potentially be augmented by the precision medicine approach in the future.
Nevertheless this latter argumentation was not included in the MS discussion, yet it is a complementary part of our answer to the reviewer here. )
We are grateful for the Reviewer for the positive comments and also for the critical questions and we do hope that our detailed answers point-by-point might be found satisfactory and this MS would be of interest to the Readers of the International Journal of Molecular Sciences.
References:
1. Beaumont, R. N.; Warrington, N. M.; Cavadino, A.; Tyrrell, J.; Nodzenski, M.; Horikoshi, M. Genome-wide association study of offspring birth weight in 86 577 women identifies five novel loci and highlights maternal genetic effects that are independent of fetal genetics. Hum Mol Genet 2018, 27, (4), 742-756.
2. Jacklin, P. B.; Maresh, M. J.; Patterson, C. C.; Stanley, K. P.; Dornhorst, A.; Burman-Roy, S.; Bilous, R. W., A cost-effectiveness comparison of the NICE 2015 and WHO 2013 diagnostic criteria for women with gestational diabetes with and without risk factors. BMJ Open 2017, 7, (8), e016621.
Round 2
Reviewer 1 Report
The general explanation of MTNR1B such as its function and the frequency of SNPs, and other related disease if any, would
In the result part lane120~121, please clarify in which population was the significant difference in FPG levels (p=0.041) found?
In explanation part of Table1 is unclear. What is p=0.0074 stands for? In the explaining part, it is supposed to be the p value for FPG being significantly high is BMI≥29 group, but in the table1, it is only placed in BMI less than 29. P=0.0413is also confusing because although you have mentioned that the FPG difference was not significant between AIT group and non-pharmacologically treated group in BMI ≥29, p=0.0413 is placed in both BMI groups.
The discussion is now much clearer than the previous manuscript. However, the total impression is still blur because the main point of this paper is hard to figure out. I though that the most significant finding of this paper is that in obese group, FPG cannot be a marker to distinguish the GDM patients who require future-insulin therapy from GDM patients who can be controlled only by non-pharmacological treatment; rather MTNR1B rs10830963 could be a marker for those future-insulin demands. I would suggest the discussion part to be more focused on your main point, using more direct and strong words.
Author Response
We are grateful for “Reviewer 2” for his/her additional questions and comments he/she made in this 2nd round of review. Please find our answers point-by-point as follows:
Comment 1 (Reviewer 2, round 2): “The discussion is now much clearer than the previous manuscript. However, the total impression is still blur because the main point of this paper is hard to figure out. I though that the most significant finding of this paper is that in obese group, FPG cannot be a marker to distinguish the GDM patients who require future-insulin therapy from GDM patients who can be controlled only by non-pharmacological treatment; rather MTNR1B rs10830963 could be a marker for those future-insulin demands. I would suggest the discussion part to be more focused on your main point, using more direct and strong words."
Answer 1:
We are grateful for the comment of the reviewer.
In general, the FPG value at OGTT is one of the predictors of AIT in a validated model [1].
In our analysis we were able to identify the MTNR1B rs10830963 variant as another important predictor of AIT in specific GDM cases. The association between MTNR1B rs10830963 and FPG was already confirmed, thus using only one of these predictors is required statistically.
We separated our GDM study population for two subgroups (lower BMI (<29 and="" higher="" bmi="">=29 kg/m2)) and observed different effects for FPG as a marker.
When the FPG and the genotype data were analysed together to predict the need for AIT, we used a two variable logistic regression model to assess whether the MTNR1B genotype or the FPG level was a more significant predictor of AIT.
We found that in the low pre-pregnancy BMI subgroup only the FPG level could be a significant marker (p=0.01), however in the higher pre-pregnancy BMI subgroup the FPG level became insignificant, in contrast carrying the G risk allele of MTNR1B rs10830963 was a significant marker (p=0.01) as indicated in the 2.3.1 section.
We inserted the following text to the Discussion (3.6. Summary):
“In general, the FPG value at OGTT was one of the predictors of AIT in a validated model [1]. In this analysis we found that in the lower pre-pregnancy BMI subgroup the FPG level was a more significant predictor of AIT than the MTNR1B genotype, however in the higher pre-pregnancy BMI subgroup the MTNR1B rs10830963 genotype became superior compared to the FPG. This indicates that the MTNR1B rs10830963 gene variant may have an additional direct effect on the initiation of AIT in the subgroup of GDM patients with pre-pregnancy BMI ≥ 29kg/m2. In addition, from the clinical perspective it is notable that the intra-individual reproducibility of PG results at 75g OGTT, including FPG are under intense scrutiny (in the non-pregnant population) [2], in contrast to the genomic DNA information that is expectedly a stable marker.”
In addition, we also made a few minor revisions [e.g.: “diet” has been changed to “MNT (Medical nutrition therapy)”] in the text and further polished the MS from the point of English language and made the other sections of Discussion also more consistent and shorter.
Comment 2: "In the result part lane120~121, please clarify in which population was the significant difference in FPG levels (p=0.041) found? In explanation part of Table1 is unclear. What is p=0.0074 stands for? In the explaining part, it is supposed to be the p value for FPG being significantly high is BMI≥29 group, but in the table1, it is only placed in BMI less than 29. P=0.0413is also confusing because although you have mentioned that the FPG difference was not significant between AIT group and non-pharmacologically treated group in BMI ≥29, p=0.0413 is placed in both BMI groups.”
Answer 2:
We have carefully revised this part of the Results section, including Table 1 (using the ANOVA with Tukey post-hoc test for the FPG data according to pre-pregnancy BMI-stratified subgroups). According to the suggestion of the Reviewer we restructured the reporting of the data in a more comprehensive manner in the MS.
We are grateful for the Reviewer for his/her questions and we do hope that our detailed answers point-by-point might be found satisfactory and improved the MS that in this revised form would be of interest to the Readers of the International Journal of Molecular Sciences.
Sincerely,
Gabor Firneisz, M.D., Ph.D.
corresponding author
References:
1. Barnes, R. A.; Wong, T.; Ross, G. P.; Jalaludin, B. B.; Wong, V. W.; Smart, C. E.; Collins, C. E.; MacDonald-Wicks, L.; Flack, J. R., A novel validated model for the prediction of insulin therapy initiation and adverse perinatal outcomes in women with gestational diabetes mellitus. Diabetologia 2016, 59, (11), 2331-2338.
2. Perry, R. C.; Shankar, R. R.; Fineberg, N.; McGill, J.; Baron, A. D.; Early Diabetes Intervention, P., HbA1c measurement improves the detection of type 2 diabetes in high-risk individuals with nondiagnostic levels of fasting plasma glucose: the Early Diabetes Intervention Program (EDIP). Diabetes Care 2001, 24, (3), 465-71.
